# Enabling Unsupervised Neural Machine Translation with Word-level Visual Representations

**Chengpeng Fu[1,2], Xiaocheng Feng[1,2], Yichong Huang[1], Wenshuai Huo[1,2],**
**Hui Wang[2], Bin Qin[1,2], Ting Liu[1,2]**
[1]Harbin Institution of Technology, Harbin, China
[2]Pengcheng Laboratory, Shenzhen, China
fuchp@stu.hit.edu.cn, {huowsh,wangh06}@pcl.ac.cn
{xcfeng,ychuang,qinb,tliu}@ir.hit.edu.cn

## Abstract

Unsupervised neural machine translation has recently made remarkable strides, achieving impressive results with the exclusive use of monolingual corpora. Nonetheless, these methods still exhibit fundamental flaws, such as confusing similar words. A straightforward remedy to rectify this drawback is to employ bilingual dictionaries, however, high-quality bilingual dictionaries can be costly to obtain. To overcome this limitation, we propose a method that incorporates images at the word level to augment the lexical mappings. Specifically, our method inserts visual representations into the model, modifying the corresponding embedding layer information. Besides, a visible matrix is adopted to isolate the impact of images on other unrelated words. Experiments on the Multi30k dataset with over 300,000 self-collected images validate the effectiveness in generating more accurate word translation, achieving an improvement of up to +2.81 BLEU score, which is comparable or even superior to using bilingual dictionaries.[1]

## 1 Introduction

Unsupervised Neural Machine Translation (UNMT) (Zhang et al., 2017; Artetxe et al., 2018; Lample et al., 2018a,b,c) seeks to avoid the necessity for high-quality bilingual data in conventional machine translation by exploring the possibility of completing translations relying solely on the monolingual corpus. Consisting of three main components, cross-lingual language model initialization, denoising auto-encoder, and iterative back translation, recent methods (Conneau and Lample, 2019; Song et al., 2019) have achieved promising results.

Nonetheless, recent UNMT approaches have been observed to make particular errors in comparison to supervised machine translation, such

as confusing nouns that pertain to the same semantic category. For instance, Bapna et al. (2022) demonstrates that their models translate "lion" to "rabbit" or "snake" despite getting good CHRF scores. In the context of supervised machine translation, pinpointing accurate translations of words is straightforward with aligned bilingual data. However, with only monolingual data, the unsupervised model must rely on large amounts of context for word mapping, which can lead to confusion between words with similar distributions, particularly when there is insufficient data. To address this issue, a direct solution is to integrate high-quality bilingual dictionaries into the UNMT process using codeswitching or spliced methods (Jones et al., 2023). High-quality bilingual dictionaries can be obtained through manual annotation and bilingual data extraction, yet these methods either involve high costs or deviate from the original objective of the UNMT task.

In addition to utilizing explicit bilingual dictionaries for word mapping, implicit methods can also be employed. As the saying goes, "A picture is worth a thousand words". We can leverage the "thousand words" represented by corresponding images to learn the mapping between words in different languages. Visual content has the potential to enhance word mapping abilities in latent spaces, as the physical visual perception is congruent among speakers of diverse languages (Huang et al., 2020). Furthermore, the cost of annotating a bilingual dictionary is relatively high, whereas the cost of searching images with monolingual keywords is relatively low in the era of sophisticated search engines. Describing an image is also less demanding for humans compared to translating a word, as the former only requires mastery of one language while the latter requires at least two.

Drawing on these insights, we propose to integrate word-level image information into the UNMT process to mitigate the issue of lexical confusion.

---

[1]The word-image dataset and associated codes are publicly available at: https://github.com/Brianmax99/UNMT-WVR

It's worth noting that we don't use sentence-level images like some multi-modal UNMT approaches (Su et al. (2019); Huang et al. (2020)). Instead, we focused on word-level images since they are more effective and flexible in addressing the challenge of lexical confusion and are easier to collect. Our approach involves augmenting the encoded images to their corresponding words, modifying the corresponding position and language encoding information, and utilizing a visible matrix to isolate the impact of images on other words. We conduct experiments utilizing over 300,000 self-collected images, and further analysis demonstrates that we have effectively mitigated lexical confusion to a certain extent. Furthermore, we achieve new state-of-the-art results for the UNMT task in certain language directions on the Multi30k dataset. The main contributions of this work can be summarized as follows:

- We introduce a concise yet potent framework that integrates word-level images to alleviate the similar lexical confusion issue in UNMT, without utilizing bilingual dictionaries.

- Our approach has achieved promising or even unparalleled unsupervised results on the Multi30K dataset for certain language pairs.

- We release the word-image datasets for five languages as open-source, which can serve as a valuable resource to the community.

## 2 Methods

In this section, we first review the text-only UNMT process. Then, we present our proposed method for incorporating images, which involves the modified embedding layer and mask transformer. Finally, we introduce some training strategies.

### 2.1 Unsupervised Neural Machine Translation

Successful UNMT systems typically employ an encoder-decoder architecture similar to that of supervised machine translation, sharing parameters between the source-to-target and target-to-source models. They share several common principles: initialization, iterative back-translation, and denoising autoencoder. Let $X = \{X_i\}_{i=1}^M$ denote monolingual data in language $L_1$ and $Y = \{Y_i\}_{i=1}^N$ in language $L_2$. $M$ and $N$ are the sentence number of monolingual data $X$ and $Y$ respectively. $\theta$ represents the UNMT model parameters.

**Initialization.** Performing parameter initialization that is non-random can introduce cross-lingual prior knowledge into the model. There are currently two main initialization methods for UNMT models. The first method initializes the embedding layer of a UNMT model with word pretrained embeddings, while the second method uses a pretrained cross-lingual language model with the same structure as the UNMT encoder to initialize most of the neural network parameters.

**Iterative Back-translation.** Back-translation is a method that leverages monolingual data in the target language and a reverse translation model to generate pseudo-source data, improving the quality of translations in the forward direction. In UNMT, where only monolingual data is available, back-translation is used to generate pseudo-parallel data, transforming the unsupervised problem into a weakly or self-supervised one. During training, back-translation is repeatedly applied in an iterative manner. The loss is defined as followed:

$$\mathcal{L}_B = \sum_{i=1}^M -log P_{L_2 \to L_1}(X_i | S_{L_1 \to L_2}(X_i, \theta), \theta) \\ + \sum_{j=1}^N -log P_{L_1 \to L_2}(Y_j | S_{L_2 \to L_1}(Y_j, \theta), \theta) \quad (1)$$

where $P_{L_a \to L_b}$ denotes the translation probabilities between two languages and $S_{L_a \to L_b}$ denotes the translation processes from language $L_a$ to $L_b$.

**Denoising Autoencoder.** The introduction of the denoising autoencoder aims to guide the model in producing fluent text output. This is achieved by adding noise to sentences through techniques such as replacement, deletion, and shuffling, and then allowing the model to output the correct and fluent original sentence. The loss is defined as follows:

$$\mathcal{L}_D = \sum_{i=1}^M -log P_{L_1 \to L_1}(X_i | \mathcal{N}(X_i), \theta) \\ + \sum_{j=1}^N -log P_{L_2 \to L_2}(Y_j | \mathcal{N}(Y_j), \theta) \quad (2)$$

where $\mathcal{N}(\cdot)$ refers to the noise functions. $P_{L_a \to L_a}$ denotes the reconstruction probabilities in $L_a$.

The final objective function of UNMT is:

$$\mathcal{L} = \mathcal{L}_B + \lambda_D \mathcal{L}_D \quad (3)$$

where $\lambda_D$ is a hyper-parameter that weights denoise autoencoder loss term and is gradually reduced during training.

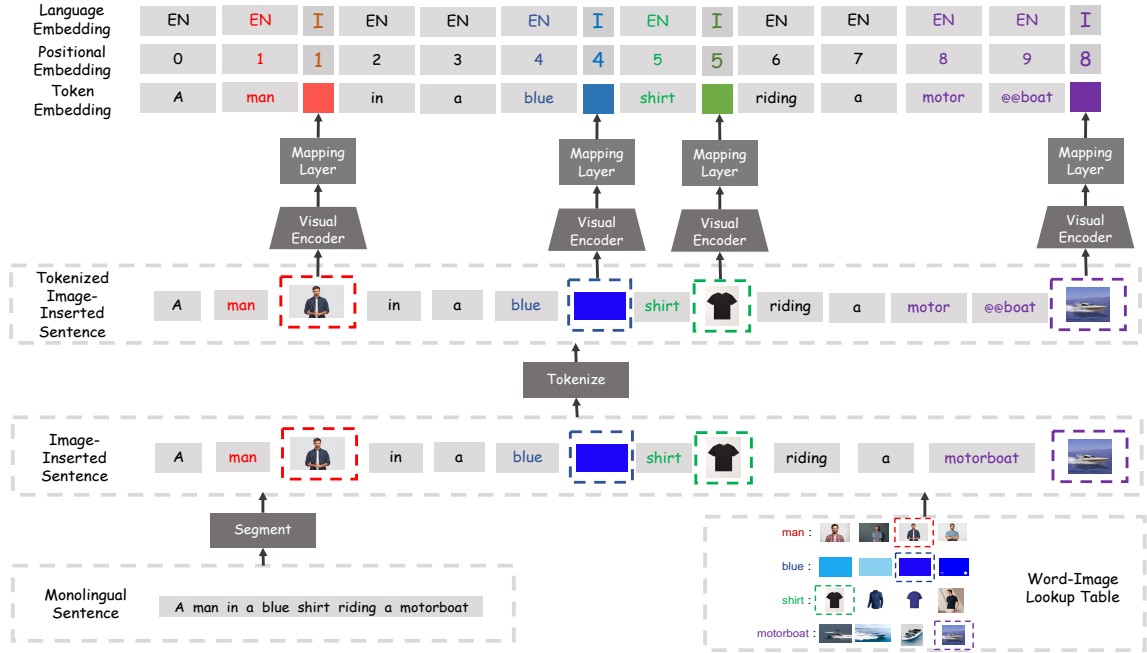

Figure 1: The process of converting an image-inserted sentence into an embedding representation.

## 2.2 Image Fusion

In this section, we present how to incorporate word-level images into the UNMT process to tackle the issue of lexical confusion. The main concept behind our approach is that words can acquire information not only from contextual information but also from their corresponding images. Let $X_i = \{x_i^1, x_i^2, ..., x_i^K\}$ denote a sentence from $X$ in language $L_1$ and $x_i^K$ denotes the $k$-th word in sentence $X_i$. We denote a lookup table consisting of word-image pairs as $\mathcal{D}_{L_i} = \{(word, I_{word})\}$, where $I_{word}$ is an image set with one or multiple images that depict the meaning of the word.

As shown in Figure 1[2], we segment an input sentence $X_i$ at the word level and obtain the images by retrieving the words appearing in the lookup table $\mathcal{D}_{L_i}$. Then, we concatenate the retrieved images to the corresponding words, resulting in an Image-Inserted Sentence denoted $IIS_i$. Each word can be concatenated with one or multiple images. Formally, We denote $IIS_i$ as follow:

$$IIS_i = \{x_i^1, \widehat{I}_{x_i^1}, x_i^2, \widehat{I}_{x_i^2}, ..., x_i^K, \widehat{I}_{x_i^K}\} \quad (4)$$

where $\widehat{I}_{x_i^K}$ is the image subset of $I_{x_i^K}$ ($\widehat{I}_{x_i^K}$ can be an empty set). We define this process of injecting images as:

$$IIS = \mathcal{I}(X) \quad (5)$$

where $IIS = \{IIS_i\}_{i=1}^M$.

After obtaining the $IIS$, we need to integrate them into the UNMT model. To achieve this integration, we employ a modified embedding layer and a mask transformer.

### 2.2.1 Modified Embedding Layer

The first modification part is the embedding layer, which generally includes token embedding, position embedding, and, in some models, language embedding. An example can be seen in Figure 1.

**Token Embedding.** In the context of $IIS$, both textual words and images are present. Given that most current machine translation methods operate at the subword level, we proceed to tokenize the $IIS$, focusing solely on the textual component of $IIS$. For encoding textual tokens, we continue to utilize the embedding layer from the original text-only model. Regarding images, we leverage the CLIP (Radford et al., 2021) model as the image encoder to extract visual features. The CLIP model is a multi-modal model trained on a variety of (image, text) pairs, which possesses a powerful capability to represent images. To prevent negative impacts on the encoding of images during the UNMT process, we freeze the visual encoder. However, directly using these representations may lead

---

[2]"Blue" is an adjective in the figure. In our image collection process, "blue" can also appear as a noun in other training samples.

to a gap between the image encoding space and the semantic space in UNMT. Therefore, we employ an MLP layer to bridge this gap.

**Position Embedding.** Positional encoding is a crucial module that supplements token structural information. For textual information in $IIS$, we continue to encode the position in the original order of the input sentence to prevent interference with sentence structure learning when integrating images into the model. For images, we assign the positional encoding of the first subword in the tokenized result for the corresponding word.

**Language Embedding.** Some UNMT models incorporate language embedding to inform the model of the input language and the desired output language. For textual information, we continue to utilize the original language encoding. However, for image information, as it does not belong to any language, we introduce a new image identifier.

### 2.2.2 Mask Transformer

Despite modifying the original embedding layer to allow $IIS$ to be properly fed into the model, two issues still exist. Firstly, inserting images may affect other irrelevant words in the original sentence and potentially disrupt the structural relationships between them. Secondly, a single word-level image may correspond to multiple subwords, and during the annotation of position embedding, only the information of the first subword is utilized, resulting in a lack of relationship between the image and the remaining subwords it corresponds to. To address these issues, we introduce the method of mask transformer (Liu et al., 2020).

We create a visible matrix to shield the relationship between images and other textual information in the sentence and highlight the relationship between the image and its corresponding subwords. The visible matrix $M$ is defined as:

$$M_{ij} = \begin{cases} -\infty & w_p \bullet w_q \\ 0 & w_p \circ w_q. \end{cases} \quad (6)$$

where $w_p$ and $w_q$ are elements in $IIS_i$, $w_p \circ w_q$ denotes that $w_p$ and $w_q$ are visible to each other, while $w_p \bullet w_q$ is not. An example of a visible matrix is depicted in Figure 2.

The visible matrix can be added to the self-attention module to mask certain attention calculations. specifically, the new masked self-attention is:

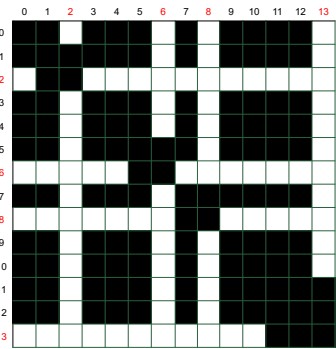

Figure 2: An example of the visible matrix corresponding to the $IIS$ in Figure 1. The red numerical labels indicate the absolute position of the images in the $IIS$. In this matrix, black represents "visible" and white represents "invisible".

$$S^i = softmax(\frac{Q^i K^{iT} + M}{\sqrt{d}}) \quad (7)$$

where $Q$ and $K$ are the query matrix and key matrix in the self-attention module. if two elements are invisible, the $M$ will mask the attention score to 0, which means making no contribution to each other.

### 2.3 Training Strategy

After initialization by cross-lingual pretrained language model, we train denoising autoencoder and iterative back-translation alternatively. The final loss of image fusion UNMT is:

$$\mathcal{L}^I = \mathcal{L}_B^I + \lambda_D \mathcal{L}_D^I \quad (8)$$

specifically, $\mathcal{L}_B^I$ and $\mathcal{L}_D^I$ refer to:

$$\mathcal{L}_B^I = \sum_{i=1}^{M} -logP_{L_2 \to L_1}(X_i | \mathcal{I}(S_{L_1 \to L_2}(\mathcal{I}(X_i), \theta)), \theta)$$
$$+ \sum_{j=1}^{N} -logP_{L_1 \to L_2}(Y_j | \mathcal{I}(S_{L_2 \to L_1}(\mathcal{I}(Y_j), \theta)), \theta)$$
$$(9)$$

$$\mathcal{L}_D^I = \sum_{i=1}^{M} -logP_{L_1 \to L_1}(X_i | \mathcal{I}(N(X_i)), \theta)$$
$$+ \sum_{j=1}^{N} -logP_{L_2 \to L_2}(Y_j | \mathcal{I}(N(Y_j)), \theta)$$
$$(10)$$

During the denoising autoencoder process, we first add noise to the original sentence and then integrate the images. During the iterative back-translation process, for the first round of synthetic data generation, we do not add image information to achieve a smoother transition because the pretrained language model did not include images.

| Language | #Word | #Image | Average |
|----------|-------|--------|---------|
| En | 4,975 | 45,228 | 9.09 |
| Fr | 4,646 | 36,927 | 7.95 |
| De | 10,384 | 85,419 | 8.23 |
| Cs | 10,706 | 72,942 | 6.81 |
| Zh | 10,805 | 85,144 | 7.88 |
| Total | 41,516 | 325,660 | 7.84 |

Table 1: The statistics of our constructed Word-Image dataset.

## 3 Experiments

In this section, we provide a detailed description of the datasets and experimental setup employed in our study. Subsequently, we perform an in-depth analysis of the results obtained.

### 3.1 Datasets

**Word-Image Dataset.** In our approach, the most crucial component of the experiment data is the image dataset at the word level. Since there is no off-the-shelf word-level image dataset, we build one for performance evaluation. We collect these images through the following steps: word extraction, image crawling, and image filtering. Firstly, extract the nouns from the monolingual dataset using part-of-speech analysis tools including Spacy[3] and Stanford CoreNLP[4]. We choose nouns because they typically correspond to real-world entities and are more likely to yield high-quality images. Next, use Google Images[5] to crawl images for these nouns, limiting the image size to between 200-by-200 and 10,000-by-10,000 pixels and retrieving 10 to 15 images per noun. The image format is restricted to JPG, JPEG, and PNG. Finally, filter the crawled images. We consider all the images collected for a single noun as a cluster and encode them using the CLIP model. We then calculate pairwise similarities between all images and remove those with an average similarity below a threshold $\mathcal{T}$, which we set to 0.6, as they are considered anomalous. The statistics information of the Word-Image dataset can be seen in Table 1. Our dataset covers a wide range of common nouns and can be reused for various purposes.

---

[3]https://github.com/explosion/spaCy
[4]https://github.com/stanfordnlp/CoreNLP
[5]https://images.google.com

**Monolingual Dataset.** The monolingual datasets used in our experiments are Multi30k dataset (Elliott et al., 2016, 2017; Barrault et al., 2018), LDC dataset and WMT News Crawl dataset. The Multi30k dataset contains images and their captions in four languages: English(En), French(Fr), Germany(De), and Czech(Cs). Every language has 29000 sentences for training and 1014 for validation. It is worth noting that we only utilized the lingual data from Multi30k as the images are sentence-level. To prevent the model from encountering bilingual corpora, we randomly split the training and validation sets of the Multi30k dataset into two equal parts. Each part of the training set contains 14,500 sentences, and each part of the validation set contains 507 sentences. We report results in all test sets in Multi30k including Test2016, Test2017, Test2018, and MsCOCO. For the Cs-En task, only Test2016 and Test2018 are available. Furthermore, we expanded the dataset by incorporating monolingual data from WMT 2008-2022 News Crawl and LDC. We also report our results on the common news domain test set, which includes WMT Newstest 2014 for En-Fr, Newstest 2021 for En-Cs, and NIST06 for En-Zh.

### 3.2 Experimental Details

**Baseline Models.** We compare recent unsupervised text-only and multimodal MT baselines listed in the following:(1) XLM (Conneau and Lample, 2019) uses a masked language model to train a cross-lingual language model to initialize. (2) UMMT (Su et al., 2019) uses visual features for denoising autoencoder and back-translation. (3) PVP (Huang et al., 2020) employs multimodal back-translation and features pseudo visual pivoting. (4) UNMT-CS (Jones et al., 2023) uses the codeswitching method, where words in the source sentence are swapped out for their corresponding translation from bilingual dictionaries, to solve the lexical confusion issue in UNMT. It is worth noting that both UMMT and PVP datasets utilize sentence-level images.

**System Settings.** We conducted our experiments based on the XLM codebase[6]. The model architecture employed is a 6-layers, 8-heads transformer, with a hidden layer representation dimension of 1024. The epoch size is 100K and the tokens-per-batch is 2000. we use Adam optimizer and 2000 warm-up updates. The learning rate is 1e-5. For

---

[6]https://github.com/facebookresearch/XLM

| Approach | En-Fr | | | | En-De | | | | En-Cs | |
|---|---|---|---|---|---|---|---|---|---|---|
| | F16 | F17 | F18 | C17 | F16 | F17 | F18 | C17 | F16 | F18 |
| UMMT | 39.80* | – | – | – | 23.50* | – | – | – | – | – |
| PVP | 52.30* | – | – | – | **33.90*** | – | **–** | – | – | – |
| XLM | 53.08 | 46.82 | 35.31 | 43.09 | 31.81 | 26.83 | 25.62 | 22.95 | 25.14 | 20.15 |
| UNMT-CS | **54.24** | 46.30 | 36.06 | 43.49 | 33.43 | 27.51 | 26.67 | 24.27 | **26.46** | 22.71 |
| Ours | 54.07 | **46.95** | **36.21** | **44.44** | 33.06 | **29.12** | **27.37** | **24.75** | 26.34 | **22.88** |
| Δ | +0.99 | +0.13 | +0.90 | +1.35 | +1.25 | +2.29 | +1.75 | +1.80 | +1.20 | +2.73 |

| Approach | Fr-En | | | | De-En | | | | Cs-En | |
|---|---|---|---|---|---|---|---|---|---|---|
| | F16 | F17 | F18 | C17 | F16 | F17 | F18 | C17 | F16 | F18 |
| UMMT | 40.50* | – | – | – | 26.40* | – | – | – | – | – |
| PVP | 46.00* | – | – | – | 31.60* | – | – | – | – | – |
| XLM | 48.04 | 42.14 | 37.03 | 43.58 | 37.51 | 33.21 | 30.73 | 27.98 | 33.13 | 30.29 |
| UNMT-CS | **49.28** | 42.86 | 37.19 | 44.39 | 38.83 | 34.48 | 31.44 | 28.07 | 33.91 | 31.33 |
| Ours | 49.10 | **43.01** | **37.41** | **44.57** | **40.32** | **35.20** | **32.70** | **29.81** | **34.29** | **31.56** |
| Δ | +1.06 | +0.87 | +0.38 | +0.99 | +2.81 | +1.99 | +1.97 | +1.83 | +1.16 | +1.27 |

Table 2: BLEU scores on Multi30k En-Fr, En-De, and En-Cs UNMT tasks. F18, F17, F16, and C17 respectively denote the Flickr2018, Flickr2017, Flickr2016, and COCO2017 test sets in Multi30k. * is the results reported from previous papers. Δ represents the improvement of our method over the XLM method in terms of BLEU.

evaluation, we use 4-gram BLEU (Papineni et al., 2002) scores by multi-bleu.pl in Moses[7]. We train all models on 4 NVIDIA 32GB V100 GPUs.

During training, we first train the models on a 5M WMT News Crawl dataset for 70 epochs and subsequently continued training on the Multi30k dataset for another 70 epochs due to its relatively small size. As for the CLIP model, we use ViT-L/14@336px. When incorporating images, we randomly select two images from the corresponding image set for each word and concatenate them to the text. We only incorporate images during the iterative back-translation and denoising autoencoder stages, so our method can be used in conjunction with other pre-training enhancement methods. Regarding the cross-lingual models used for initialization, we directly employ high-quality open-source cross-lingual models in experiments involving En-Fr, En-De, and En-Zh. However, for the En-Cs, we train a model using WMT monolingual dataset, with 10 million English sentences and 10 million Czech sentences, due to the unavailability of any open-source models.

### 3.3 Main Results

Table 2 shows the main results on the Multi30k test set, including translating among English, French, German, and Czech. Comparing the results of XLM and our method, we observe that adding im-

ages leads to performance improvements in all six translation directions, with a maximum increase of +2.81 BLEU points, indicating the effectiveness of our approach. UNMT-CS is a method that utilizes bilingual dictionaries to address the problem of lexical confusion via code-switching. By comparing our method with the UNMT-CS approach, we observed comparable results, and in some directions, our method even outperformed the code-switching approach, indicating that ours is more efficient and cost-effective.

When compared to the best-performing method that incorporates sentence-level images, our approach still achieves comparable or better results in most translation directions. This is despite the matching degree between the Multi30k monolingual data and word-level images being slightly lower than that of sentence-level images, as the text data in Multi30k is generated based on the sentence-level images. In some directions, our approach even outperforms the previous best methods by up to +4.22 BLEU points (as shown in Table 2, where our method is compared to the first two methods of the table).

## 4 Analysis

### 4.1 Case Study

In Table 3, we provide several examples of translations, where we observe that the XLM method re-

[7] https://github.com/moses-smt/mosesdecoder

| | |
|---|---|
| GT-DE | Menschen an der Seitenlinie bei einem Fußballspiel. |
| GT-EN | People on the sideline of a soccer match. |
| XLM | People at the plate in a baseball game. |
| Ours | People at the stands in a soccer game. |
| GT-DE | Ein kleines Kind in einer orangefarbenen Rettungsweste hält ein Paddel und paddelt in einem blauen Kajak auf einem Gewässer. |
| GT-EN | A young child wearing an orange life vest holding an oar paddling a blue kayak in a body of water. |
| XLM | A small child in an orange-red life vest holds a Paddy Power and paddles in a blue cadet on a body of water. |
| Ours | A small child in an orange life vest is holding a paddle and paddling in a blue kayak on a body of water. |
| GT-FR | Un homme avec une veste noire, une casquette à carreaux et un pantalon rayé noir et blanc joue de la guitare électrique sur une scène avec un chanteur et un autre guitariste en arrière-plan. |
| GT-EN | A man in a black jacket and checkered hat wearing black and white striped pants plays an electric guitar on a stage with a singer and another guitar player in the background. |
| XLM | A man with a black jacket, striped headband and light brown and white shirt plays an electric guitar on a stage with a singer and another guitarist in the background. |
| Ours | A man with a black jacket, checkered cap and black and white striped pants plays an electric guitar on a stage with a singer and another guitarist in the background. |

Table 3: Qualitative results of the proposed model. GT denotes ground truth. Words highlighted in red are affected by the problem of lexical confusion.

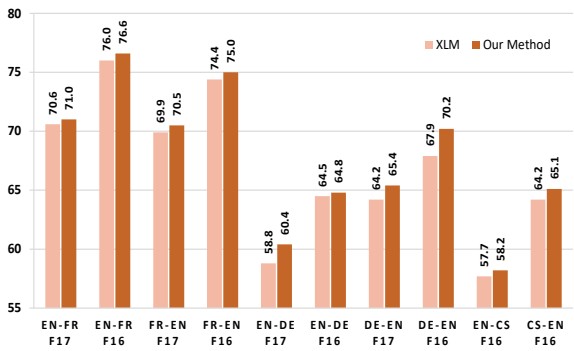

Figure 3: 1-gram precision scores on Flickr2016 and Flickr2017 data sets.

sults in some lexical confusion errors. For instance, in Example 1, "Fußballspiel" is translated to "baseball" instead of "soccer", and in Example 2, "orangefarbenen" is translated to "orange-red" instead of "orange". The representation of these words is similar, which makes it difficult for the model to differentiate between them. However, when we incorporate image information, the words obtain additional information from the image representation, which helps alleviate this problem to some extent. In Table 3, we can see that these words are correctly translated when we add image information. Therefore, our approach can effectively address the issue of lexical confusion in UNMT.

### 4.2 1-gram Precision Scores

To better observe how our method can alleviate the issue of lexical confusion, we report the 1-gram precision scores in Figure 3. The accuracy of the model's word translations can be reflected intuitively in the 1-gram precision scores metric. We report the results of our approach with and without

incorporating images for various translation directions on the Flickr2016 and Flickr2017 test sets and find that our method with image incorporation outperforms the method without images in all directions. This demonstrates that our method can indeed provide more accurate word translations.

### 4.3 Performance on WMT and LDC Dataset

Not all sentences have a corresponding image that perfectly matches the text. On the other hand, generally, each sentence has some words that can be matched with images, making the use of word-level images more flexible and less costly to collect. Hence, we also conduct experiments on a more common news dataset. We conduct experiments on En-Fr, En-Cs, and En-Zh translation directions using the WMT 5M monolingual data for En-Fr and En-Cs, and the LDC dataset for En-Zh. We split the LDC bilingual data into two non-overlapping parts, each containing 0.75 million monolingual sentences. The news dataset is more abstract in content compared to the Multi30k dataset, and it is difficult to find an image that perfectly matches a sentence. This makes it challenging or almost impossible to apply previous methods that rely on sentence-level images to this dataset. As shown in Table 4, our method can be applied to the news dataset, and it shows improvements compared to the method that only uses text. The improvement on the WMT dataset was not significant, and we speculate that this may be due to the sufficient size of the WMT dataset and the lower frequency of noun entities with images compared to Multi30k.

| Approach | Zh-En | En-Zh | Cs-En | En-Cs | Fr-En | En-Fr |
|---|---|---|---|---|---|---|
| XLM | 8.43 | 7.91 | 25.14 | 17.79 | 33.73 | 35.77 |
| Our Method | **9.87** | **9.36** | **25.51** | **18.24** | **34.14** | **36.46** |
| Δ | +1.44 | +1.45 | +0.37 | +0.45 | +0.41 | +0.69 |

Table 4: BLUE scores on WMT and LDC test sets.

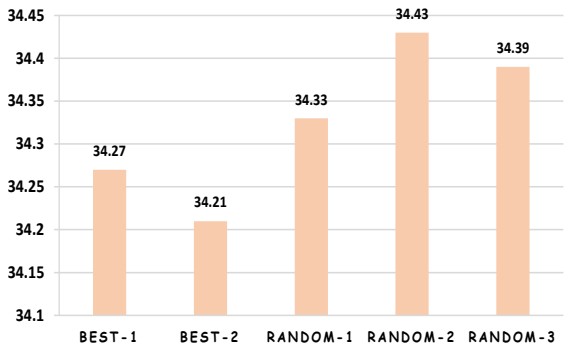

Figure 4: BLEU scores on En-De with different ways and numbers of adding image.

## 4.4 Quantity of Images

We are also curious about how the method of adding images and image quantity affect the performance. Therefore, we test several ways of adding images, including selecting the best one, the top two, randomly selecting one, randomly selecting two, and randomly selecting three. We treat all images corresponding to a single word as a class and selected the image with the highest average similarity to the other images in the same class as the best image. We report the average BLEU score of En-De on the Flickr2016 and Flickr2017 test sets as illustrated in figure 4. We find that the fixed-best method yielded lower performance compared to randomly adding images. This could be due to the limitation of the fixed-best method in capturing the full range of visual variations that may exist for a given word. Among all the random selection methods, we find that adding two images randomly yielded the best performance. This could be because adding three images may introduce more noise, while adding only one image may not provide enough randomness.

## 5 Related Work

**Unsupervised Machine Translation.** Unlike common machine translation (Sutskever et al., 2014; Vaswani et al., 2017; Feng et al., 2020), UNMT don't use parallel corpus. Recent advancements (Zhang et al., 2017; Artetxe et al., 2018; Lample et al., 2018a; Conneau and Lample, 2019;

Song et al., 2019; Han et al., 2021) in UNMT have achieved remarkable milestones. However, there are still certain issues associated with solely utilizing monolingual data for translation, such as domain mismatch problem (Marchisio et al., 2020; Kim et al., 2020), poor performance on distance language pairs (Kim et al., 2020; Chronopoulou et al., 2020; Sun et al., 2021), translationese (He et al., 2022), and lexical confusion (Bapna et al., 2022; Jones et al., 2023). Our work aims to address the issue of lexical confusion by utilizing word-level image data, thereby eliminating the need for costly bilingual dictionary annotations.

**Multimodal Machine Translation.** Multimodal machine translation involves incorporating image information into a machine translation model to enhance its performance. While most current multimodal machine translation systems (Caglayan et al., 2016; Yao and Wan, 2020; Yin et al., 2020; Caglayan et al., 2021; Zhang et al., 2020; Fang and Feng, 2022; Li et al., 2022) focus on supervised machine translation, it has been observed that visual information is generally ignored when parallel corpora are plentiful (Caglayan et al., 2019; Wu et al., 2021). Yang et al. (2022) proposed the use of image information in situations where parallel corpora are not abundant. On the other hand, Su et al. (2019) and Huang et al. (2020) utilized image information in an unsupervised setting. However, these approaches were based on sentence-level image information, which can be challenging to obtain since not all sentences have corresponding images. In contrast, word-level image information is easier to obtain and is more flexible and lightweight. Therefore, our work aims to use word-level image information to enhance the performance of UNMT.

## 6 Conclusion and Future Work

In this paper, we propose a method to alleviate the problem of lexical confusion in unsupervised machine translation by utilizing word-level images and provide a concise yet potent framework for integrating them into an unsupervised neural machine translation model. Through extensive experiments, we demonstrate that our method can effectively mitigate the issue of word confusion and even outperform the use of costly bilingual dictionaries in some directions. Furthermore, we release an open-source word-image dataset that covers 5 languages and involves over 300,000 images in total. In the future, we will endeavor to explore the

potential of utilizing images for augmenting the translation quality of low-resource languages in a large-language-model-based translation system.

## Limitations

One limitation of our work is that we have not been able to effectively handle the issue of polysemy. The word-image dataset is obtained through direct web scraping of words, and the images collected may not all correspond to the same sense of the word. Moreover, when adding images, we have not been able to accurately select images that correspond to the intended meaning of the word in the sentence. However, we have attempted to alleviate this issue to some extent by randomly selecting images during the process of adding images, which has enabled the model to encounter the correct images to a certain degree. In the future, we plan to improve our approach by incorporating sentence-level information when inserting images.

## Ethics Statement

In our experimental endeavors, we have amassed a substantial collection of images, which naturally prompts discussions pertaining to these issues. It is imperative to clarify that we do not possess the copyright for these images. Instead, they will be exclusively available to researchers and educators seeking to utilize the dataset for non-commercial research and/or educational purposes.

## Acknowledgements

Xiaocheng Feng is the corresponding author of this work. We thank the anonymous reviewers for their insightful comments. This work was supported by the National Key RD Program of China via grant No. 2021ZD0112905, National Natural Science Foundation of China (NSFC) via grant 62276078, the Key RD Program of Heilongjiang via grant 2022ZX01A32, the International Cooperation Project of PCLPCL2022D01 and the Fundamental Research Funds for the Central Universities (Grant No.HIT.OCEF.2023018).

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

## A The Implementation Details of the XLM and UNMT-CS Method

The XLM and UNMT-CS methods are re-implemented on the Multi30k dataset in our paper. During training, both these two methods are trained on the 5 million sentences from the WMT dataset for 70 epochs and subsequently continued training for 70 epochs on the Multi30k dataset. For the bilingual dictionary in codeswitching method, we use the Ground-truth bilingual dictionaries provided in MUSE[8] (Lample et al., 2018b). When training, we substituted words in 50% of the sentences, while the remaining sentences were left unchanged. All other parameters used in the XLM and UNMT-CS methods are identical to those used in our proposed method, as described in our paper.

## B The Implicit Mapping Relationship Between Image Representations

In fact, our approach doesn't explicitly provide the model with a mapping relationship. Rather, we introduce the (word-image) pairs relationship into the model, enabling it to extract information from similar images and thus enhancing the mapping relationship between words.

We are also curious about the impact on results if the implicit mapping relationship between image representations was absent. Therefore, during the second stage of multi30k training, we replace all images from one language with randomly generated representations while keeping the other language's images unchanged. The En→De direction results for the Flickr2016 and Flickr2017 datasets experience a significant decline, decreasing from 40.32/35.20 to 39.17/34.29. This confirms the crucial importance of the mapping relationship between images for our approach.

---

[8]https://github.com/facebookresearch/MUSE