# OpenReview forum: "Enabling Unsupervised Neural Machine Translation with Word-level Visual Representations"
_EMNLP/2023/Conference — EMNLP 2023 Findings_

### Official Review · Reviewer_Latj · 2023-07-25

**Soundness:** 3

**Excitement:**

4: Strong: This paper deepens the understanding of some phenomenon or lowers the barriers to an existing research direction.

**Paper Topic And Main Contributions:**

This paper introduces a novel multi-modal unsupervised neural machine translation method based on word-level images. The authors propose an image fusion technique in the embedding layer of the Transformer and collect a word-image dataset containing 300K images for conducting experiments. To enhance the integration of images with words in the original sentences, the authors further propose a visible matrix to shield the relationship between images and words.


**Questions For The Authors:**

1. What will happen if we replace the corresponding images with random images?

**Reasons To Accept:**

The authors propose a novel multi-modal UNMT method based on word level instead of sentence level.

**Reasons To Reject:**

- Unfair comparison to baselines: The authors employ existing high-quality pre-trained cross-lingual models, while the baselines, like PVP, train their model without using pre-trained models. Moreover, the images used in this paper are self-collected, whereas other baselines use images from Multi30K and COCO datasets.

- The absence of an ablation study to demonstrate the role of word-level images is noted. Additionally, in Section 4.4, the results of the random pick are higher than the best pick. This raises a question: what will happen if we replace the corresponding images with random images?

- The authors use XLM as the text-only baseline, however, the MASS has better performance than XLM in UNMT.

**Reproducibility:**

4: Could mostly reproduce the results, but there may be some variation because of sample variance or minor variations in their interpretation of the protocol or method.

**Reviewer Confidence:**

4: Quite sure. I tried to check the important points carefully. It's unlikely, though conceivable, that I missed something that should affect my ratings.

---

> ### Author Rebuttal · Authors · 2023-08-28
>
> Thank you for your valuable feedback and support of this work. We appreciate the opportunity to address the concerns you have raised:
>
> **a) What will happen if we replace the corresponding images with random images?**
>
> Thank you for your suggestion. We will add another ablation study to this paper. As shown in Table, we have observed a significant decline when using random images during the multi30k training stage. We will include the complete results in our paper at a later stage.
>
> | | FR-EN | FR_EN | DE-EN | DE_EN |
> |---|---|---|---|---|
> |Test Sets|F16|F17|F16|F17|
> |Corresponding Image| 49.10 | 43.01 | 40.32 | 35.20 |
> |Random Image | 48.11 | 41.69 | 39.02 | 34.38 |
>
>
> **b) Unfair comparison to baselines.**
>
> 1) All the baselines uses pre-trained models including PVP which they *[1]* introduce in section 4.3 in their paper.
> 2) We employ self-collected images due to the lack of readily available word-level image datasets. Existing datasets such as Multi30k and COCO primarily consist of sentence-level images, making them unsuitable for our specific needs. This utilization of self-collected word-level images is one of the primary contributions of our paper. By comparing our method with approaches that utilize sentence-level images, we aim to highlight the unique flexibility and advantages offered by word-level images.
>
>
> **c) Random are better than best method in Section 4.4.**
>
>
> The "Random method" refers to the process of selecting one corresponding image from multiple images associated with a word, in which each image is chosen randomly for each training sample, allowing the model to have exposure to all the images during training including the best image. However, the "best method", which involves fixing the best image, may limit the model's perspective. Therefore, it is expected that the best method would yield slightly lower results compared to the random method.
>
> **d) Use XLM not MASS as the text-only baseline.**
>
> UNMT encompasses two distinct training stages: pre-training and NMT training. In our approach, we specifically focus on the NMT training stage, leveraging word-level images. It is important to note that our contribution is independent and complementary to the MASS approach, which primarily emphasizes the pre-training stage. Although our method can be applied to MASS, for this particular study, our primary focus is on enhancing the more classical and widely utilized XLM model, which serves as our text-only baseline.
>
> **Reference**
>
> *[1] Po-Yao Huang, Junjie Hu, Xiaojun Chang, and Alexander Hauptmann. Unsupervised multimodal neural machine translation with pseudo visual pivoting. In Proceedings of the 58th Annual Meeting of the Association for Computational Linguistics, pages 8226–8237, Online, July 2020. Association for Computational Linguistics.*

---

### Official Review · Reviewer_jZHw · 2023-07-31

**Soundness:** 3

**Excitement:**

4: Strong: This paper deepens the understanding of some phenomenon or lowers the barriers to an existing research direction.

**Missing References:**

NA

**Paper Topic And Main Contributions:**

The paper is centered around using multimodality (image signal) for improving Unsupervised machine translation. It is based on the fact that the images would have same meaning irrespective of the language and could be useful in improving UNMT. Instead of a dictionary they leverage search engines to extract images that can be used for training with denoising autoencoder loss and backtranslation based loss. The paper's main contribution lies in the fact that they design the framework to integrate these image signals very effectively into the training pipeline.

**Questions For The Authors:**

While the authors have justified that mapping layers would bring the embeddings space together, I feel alignment might still not be fully present between XLM text embeddings and CLIP embeddings. I am curious to know what makes authors think that adding a single mapping layer and using it during training would be able to align these embeddings properly.

**Reasons To Accept:**

The paper is extremely well structured and conveys its ideas effectively. The experimental design has been given a thoughtful consideration for instance the use of masked transformer and the use of mapping layer before using it as token embedding to bring text and image embeddings to the same embeddings space. The proposed approach does show experimental gains that translate into increased BLEU score in comparison to the baseline. Overall, if selected some of the ideas might be borrowed from this paper which could help advance the field.

**Reasons To Reject:**

While the paper tries an interesting approach, it does have certain weaknesses based on my understanding and would love to hear back from the authors on these pointers:
(1) The paper is hugely reliant on search engine results. If the search engine results are biased, this would eventually bring that bias into their model which might raise some ethical concerns if they don't curate the images properly and if the curation is done properly the paper loses its edge on scalability over dictionary-based approaches. Also, in the image I feel "blue" is an "adjective" and not a noun which also raises the point the approach also depends on efficacy of the tools like "spacy" and "core nlp" and might need some addition of context and issues like polysemy.
(2) The selection of what images to take and how many images to take is also not fully justified and reasoning why 2 images work best did not seem satisfactory. Moreover, the improvements were not observed in WMT dataset (which authors justify saying lower frequency of nouns, my counter is in real life you would have many cases where nouns are not present) which raises some questions over the approach and need for more evidence. Also, in table 3, XLM was used as baseline to show delta imrovement in BLEU score in table 2 which to me felt like slightly misleading in first go. Shouldn't the deltas be compared from the best model in column.

**Reproducibility:**

5: Could easily reproduce the results.

**Reviewer Confidence:**

4: Quite sure. I tried to check the important points carefully. It's unlikely, though conceivable, that I missed something that should affect my ratings.

**Typos Grammar Style And Presentation Improvements:**

Maybe "Blue" example in figure is debatable if this can be treated as noun or an adjective

---

> ### Author Rebuttal · Authors · 2023-08-28
>
> Thank you for your valuable feedback and the endorsement of our research. Here is our response to your inquiries:
>
> **a) Hugely reliant on search engine results.**
> 1) The images we retrieve are based on nouns, which results in relatively fewer biases compared to other complex search.
> 2) To mitigate this issue, we utilize a clustering algorithm during the data collection process, enabling the implementation of an automatic filtering method to remove low-quality images. While this approach may seem straightforward, our results shows that appears to be sufficient.
> 3) In cases where a higher quality dataset is desired, manual selection can be employed, albeit at a higher cost. However, it is important to note that manual image selection is still less expensive compared to annotating bilingual dictionaries, as it requires proficiency in only one language, whereas the latter necessitates proficiency in two languages.
>
> **b) "Blue" in the image.**
>
> "Blue" is indeed an adjective in this image. However, in our image collection process, "blue" can also appear as a noun in other training samples, which is why it is included in our word-image lookup table. When inserting images in our fusion method, we do not differentiate between whether a word is a noun or not. We appreciate your observation and will ensure to include this clarification in the paper.
>
> **c) The selection of what images to take and how many images to take.**
>
> In section 4.4, we conduct validation experiments and find that randomly selecting two images on the Multi30k dataset yields the best results. While these methods may introduce slight performance differences, it is important to emphasize that all of these approaches contribute to addressing the issue of word confusion to some extent.
>
> **d) The results on the WMT dataset did not show significant improvements.**
>
> The motivation behind our method stems from the need to address word confusion in UNMT, which tends to be more prevalent in nouns. Consequently, when the occurrence of proper nouns is limited or when the dataset is abundant, the issue of word confusion becomes less prominent, and the effectiveness of our method may not be significant.
>
> **e) $\Delta$ represents the improvement of our method over the XLM not the best.**
>
> Our method builds upon XLM as our fundamental baseline, incorporating modifications that directly address word confusion without relying on expensive bilingual dictionaries. However, we acknowledge that the initial presentation may have been slightly misleading. We appreciate you raising this question, and we will ensure to include an additional comparison with the best results in future versions to provide a more comprehensive and accurate assessment.
>
> **f) Single mapping layer works.**
>
> In fact, previous studies such as FROMAGe *[1]*,  LLaVAR *[2]*, and VPGTrans *[3]* have successfully employed a straightforward fully connected layer to align textual and visual spaces. These works have provided compelling evidence that a simple fully connected layer is highly effective in addressing the task at hand.
>
> **Reference**
>
> *[1] Jing Yu Koh, Ruslan Salakhutdinov, and Daniel Fried. Grounding language models to images for multimodal inputs and ouputs,ICML,2023*
>
>
> *[2] Yanzhe Zhang, Ruiyi Zhang, Jiuxiang Gu, Yufan Zhou, Nedim Lipka, Diyi Yang, and Tong Sun. Llavar: Enhanced visual instruction tuning for text-rich image understanding. arXiv e-prints, pages arXiv–2306, 2023*
>
> *[3] Ao Zhang, Hao Fei, Yuan Yao, Wei Ji, Li Li, Zhiyuan Liu, and Tat-Seng Chua. Transfer visual prompt generator across llms,2023*

---

### Official Review · Reviewer_eGcr · 2023-08-02

**Soundness:** 3

**Excitement:**

3: Ambivalent: It has merits (e.g., it reports state-of-the-art results, the idea is nice), but there are key weaknesses (e.g., it describes incremental work), and it can significantly benefit from another round of revision. However, I won't object to accepting it if my co-reviewers champion it.

**Paper Topic And Main Contributions:**

This paper is about using word-level images to enhance unsupervised neural machine translation and reduce lexical confusion.
The problem this paper addresses is how to improve the performance and accuracy of UNMT, especially for words that are easily confused due to similar distributions or meanings in different languages. The main contributions that it makes towards a solution are:
+ A method that leverages the visual information of word-level images to augment the lexical mappings between languages, and a modified embedding layer and a mask transformer to fuse image and text information in the model.
+ An extensive evaluation that demonstrates the effectiveness of the proposed method in mitigating lexical confusion and achieving comparable or better results than previous methods that use sentence-level images or bilingual dictionaries.

**Questions For The Authors:**

a) How do you select the nouns for the word-image lookup table? and How do you ensure the coverage and quality of the word-image pairs?

b) How do you handle noisy or irrelevant images that may be crawled from Google Images?

c) How do you handle words that have multiple meanings or senses, and may correspond to different images depending on the context? Do you use any word sense disambiguation technique or select images dynamically based on the sentence?

d) How do you deal with named entities? Do you use any placeholder or fallback mechanism? How does this affect the performance of UNMT?

e) How do you evaluate the impact of word-level images on the latent space of UNMT? Do you use any visualization or analysis tool to show how the images affect the word representations or alignments? Do you have any examples or cases that illustrate this impact?

**Reasons To Accept:**

The strengths of this paper are:
+ It addresses a challenging and important problem in UNMT, which is lexical confusion, and proposes a novel and simple solution that uses word-level images as an implicit bilingual resource.
+ It provides a comprehensive and rigorous experimental setup, including multiple datasets, language pairs, baselines, and evaluation metrics, and shows consistent and significant improvements over text-only and multimodal baselines.
+ It introduces a large-scale word-image dataset that covers five languages and involves over 300,000 images, which can be useful for various multimodal NLP tasks and applications.

The main benefits to the NLP community if this paper were to be presented at the conference or accepted into Findings are:
+ It would inspire more research on using word-level images for NMT and other NLP tasks, such as cross-lingual image captioning, image retrieval, or visual question answering.
+ It would provide a new perspective on how to leverage visual information for NLP, especially for low-resource or distant languages, where bilingual dictionaries or parallel corpora are scarce or unavailable.
+ It would offer a valuable dataset that can facilitate multimodal NLP research and development, and potentially lead to new tasks or challenges that combine vision and language.

**Reasons To Reject:**

The weaknesses of this paper are:
+ It does not provide a thorough analysis of the impact of different types or numbers of images on the translation quality, such as how to select the most relevant or diverse images for a given word, or how to balance the trade-off between image quantity and quality.
+ It does not compare its method with other methods that use word-level images for NMT or other NLP tasks, such as Wang et al. (2019) or Huang et al. (2021), which could provide more insights into the advantages and limitations of different approaches.
+ It does not conduct experiments on more challenging language pairs or domains, such as languages with different scripts or writing systems, or domains with more abstract or specialized vocabulary, which could test the robustness and generalizability of the proposed method.

The main risks of having this paper presented at the conference (other than lack of space to present better papers) or accepted into Findings are:
+ It could raise ethical or social issues related to the use of images for NLP, such as privacy, consent, bias, or fairness, especially when dealing with sensitive or personal data, such as names, faces, locations, or identities.
+ It could encounter technical or practical challenges related to the scalability or efficiency of the proposed method, such as how to collect, store, process, or access large amounts of word-level images for different languages or domains, or how to handle noisy or irrelevant images that may degrade the performance.
+ It could face theoretical or conceptual limitations related to the use of word-level images for NLP, such as how to account for the ambiguity, variability, or context-dependency of visual information, or how to integrate other modalities or sources of information that may complement or contradict the images.

**Reproducibility:**

3: Could reproduce the results with some difficulty. The settings of parameters are underspecified or subjectively determined; the training/evaluation data are not widely available.

**Reviewer Confidence:**

3: Pretty sure, but there's a chance I missed something. Although I have a good feel for this area in general, I did not carefully check the paper's details, e.g., the math, experimental design, or novelty.

---

> ### Author Rebuttal · Authors · 2023-08-28
>
> Thank you so much for your insightful comments and affirmation of this work. Please allow us to address the issues you raise:
>
> **Reply to Questions**
>
> **a) How to select the nouns and how do you ensure the coverage.**
>
> We elaborated on these details in Section 3.1 of our paper. Specifically, we utilize tools such as Spacy to perform part-of-speech analysis on the dataset and extract nouns. By choosing all the nouns within the monolingual dataset, we can ensure a guaranteed coverage rate.
>
> **b) How do you handle noisy or irrelevant images.**
>
> We presented the filter details in Section 3.1 of our paper. We encode all the images using the CLIP model and then calculate pairwise similarities. We remove those outlier images whose average similarity with other images falls below a certain threshold. We also randomly selected a subset of the data for manual evaluation, where the relevance was found to be greater than 95%.
>
> **c) Polysemy.**
>
> We discussed polysemy in Section Limitations.
> To mitigate this issue in some degree, we randomly select images during the image fusion process, which has provided the model with exposure to the correct images to a certain extent.
> A more precise approach to address this issue is to calculate the similarities between the context-aware word embeddings and image representations via the multilingual multi-modal language models (MMLMs). However, existing MMLMs rely on bilingual parallel text, which betrays the intention of UNMT. Considering the costly endeavor of training a new MMLM from scratch without using bilingual corpora, we deem the efficient resolution of polysemy as a valuable area for future exploration.
>
> **d) Deal with named entities.**
>
> In our paper, we specifically address the issue of word confusion with simple nouns and we have not yet considered complex named entities, as these entities are less prone to confusion.
> Certainly, we can employ the same methodology for handling named entities. In this case, we would first need to identify the named entities in the text after which we can gather images at the granularity of named entities.
> We value your suggestion and will take it into consideration for future research, aiming to optimize the treatment of named entities in our methodology.
>
> **e) Visualization.**
>
> Thank you so much for your advice. We will include some visualization to evaluate the impart of word-level images on the latent space. Specifically, we will utilize the t-SNE tool to demonstrate the representation of words before and after the insertion of images.
>
> **Rebuttal to Reject Reasons**
>
> **f) Analysis of the impact of different types or numbers of images on the translation quality.**
>
> We conducted analysis experiments on different methods of selecting images and the number of images in Section 4.4. We found that randomly selecting two images on the Multi30k dataset yields favorable results. However, it is crucial to emphasize that all of these selection methods yield comparable results and make a significant contribution in addressing the issue of word confusion to some extent.
>
>
> **g) Word-level images in other NLP tasks.**
>
> In the context of UNMT, images play a significant role by acting as intermediaries that help bridge the language gap and facilitate alignment. However, in general NLP tasks, images are primarily utilized for enhancing representations. Certainly, we also desire to compare our work with relevant research in other NLP tasks. However, we couldn't locate the specific papers you mentioned. Could you please provide the accurate titles of the papers? Thank you so much.
>
> **h) Experiments on more challenging language pairs or domains.**
>
> The current unsupervised machine translation research often focuses on datasets such as WMT and Multi30k.
> In our study, we also included an additional evaluation on the EN-ZH (language pair with different scripts) using part of LDC dataset.
> As for different domains, Our approach is domain-agnostic and can be applied universally. We also test our method on WMT19 Medline EN-DE testset in following table and find that our method showed promising results.
>
> | | DE-EN | DE_EN | EN_DE | EN-DE |
> |---|---|---|---|---|
> |Test Set|Medline 18|Medline 19|Medline 18|Medline 19|
> |XLM|9.75|11.67|7.19|6.16|
> |Ours|10.55|13.87|7.79|6.95|
>
> **i) It could raise ethical or social issues related to the use of images for NLP.**
>
> In our experimental endeavors, we have amassed a substantial collection of images, which naturally prompts discussions pertaining to these issues.
> It is imperative to clarify that we do not possess the copyright for these images. Instead, they will be exclusively available to researchers and educators seeking to utilize the dataset for non-commercial research and/or educational purposes.
> When publishing the dataset, we will also strive to mitigate privacy, fairness and other concerns related to the images. For instance, we take measures to remove images containing recognizable faces to ensure privacy.
> We sincerely appreciate you highlighting this matter, and we will ensure to incorporate this information into the paper or documentation accompanying the datasets.
>
> **j) It could encounter technical or practical challenges related to the scalability or efficiency of the proposed method.**
>
> The challenges faced in these practical applications are indeed common issues in multi-modal machine translation tasks. However, our word-level images provide distinct advantages when compared to the approaches of searching or storing a suitable sentence-level image for each individual sentence. In practice, sentence-level images should be typically obtained at the dataset level, ranging from tens of thousands to millions. However, word-level images usually require a vocabulary-level size, typically in the range of thousands. Our smaller scale results in reduced costs in terms of collection, storage, and processing requirements.
>
> **k) It could face theoretical or conceptual limitations related to the use of word-level images for NLP.**
>
> While word-level images may encounter limitations in conceptual coverage, we possess distinct advantages over sentence-level images. We provide greater flexibility and are effective in addressing a significant portion of word confusion challenges. Looking ahead, our future plans will involve some integrating concepts at the conceptual level by combining multiple words.

---

### Meta-Review · Area_Chair_gSNp · 2023-09-02

**Recommendation:** 3
**Confidence:** 3

**Metareview:**

This work proposes unsupervised NMT improvement by grounding words in multiple languages to images, so that visual words representations are mixed with text embeddings. The authors show +2.81 BLEU improvement, which is comparable to using a bilingual dictionary.

Pros: The paper presents a simple and original method. The scale of the experiment and the data released to the community is valuable (300k images with captions, 5 languages).  Results show consistent improvements in translation quality.
Cons:  The experimentation and analysis of the approach are somehow limited and preliminary. There are many open questions that would need to be addressed, as pointed out by the reviewers.  Also, experimental results depend on the images retrieved by a specific search engine. While this is fine for the case of nouns covered by the paper, it is not clear how this solution scales to other words or terms. Finally, there are ethical concerns about the data set that needs to be addressed before its release.

---

### Decision · Program_Chairs · 2023-10-07

**Decision:**

Accept-Findings

**Comment:**

This work proposes unsupervised NMT improvement by grounding words in multiple languages to images, so that visual words representations are mixed with text embeddings. The authors show +2.81 BLEU improvement, which is comparable to using a bilingual dictionary.

Pros: The paper presents a simple and original method. The scale of the experiment and the data released to the community is valuable (300k images with captions, 5 languages).  Results show consistent improvements in translation quality.
Cons:  The experimentation and analysis of the approach are somehow limited and preliminary. There are many open questions that would need to be addressed, as pointed out by the reviewers.  Also, experimental results depend on the images retrieved by a specific search engine. While this is fine for the case of nouns covered by the paper, it is not clear how this solution scales to other words or terms. Finally, there are ethical concerns about the data set that needs to be addressed before its release.